# Optical Coherence Tomography Angiography Metrics Monitor Severity Progression of Diabetic Retinopathy—3-Year Longitudinal Study

**DOI:** 10.3390/jcm10112296

**Published:** 2021-05-25

**Authors:** Inês P. Marques, Sophie Kubach, Torcato Santos, Luís Mendes, Maria H. Madeira, Luis de Sisternes, Diana Tavares, Ana Rita Santos, Warren Lewis, Conceição Lobo, Mary K. Durbin, José Cunha-Vaz

**Affiliations:** 1AIBILI—Association for Innovation and Biomedical Research on Light and Image, 3000-548 Coimbra, Portugal; ipmarques@aibili.pt (I.P.M.); tsantos@aibili.pt (T.S.); lgmendes@aibili.pt (L.M.); mhmadeira@aibili.pt (M.H.M.); dstavares@aibili.pt (D.T.); asantos@aibili.pt (A.R.S.); clobo@aibili.pt (C.L.); 2Coimbra Institute for Clinical and Biomedical Research (iCBR), Faculty of Medicine, University of Coimbra, 3000-548 Coimbra, Portugal; 3Research and Development, Carl Zeiss Meditec, Dublin, CA 94568, USA; sophie.kubach@zeiss.com (S.K.); luis.desisternes@zeiss.com (L.d.S.); warren.h.lewis@gmail.com (W.L.); mary.durbin@zeiss.com (M.K.D.); 4Department of Orthoptics, School of Health, Polytechnic of Porto, 4200-072 Porto, Portugal; 5Department of Ophthalmology, Centro Hospitalar e Universitário de Coimbra (CHUC), 3000-075 Coimbra, Portugal

**Keywords:** OCTA, diabetes, retinopathy, severity, ischemia

## Abstract

To examine retinal vessel closure metrics and neurodegenerative changes occurring in the initial stages of nonproliferative diabetic retinopathy (NPDR) and severity progression in a three-year period. **Methods:** Three-year prospective longitudinal observational cohort of individuals with type 2 diabetes (T2D), one eye per person, using spectral domain-optical coherence tomography (SD-OCT) and OCT-Angiography (OCTA). Eyes were examined four times with one-year intervals. OCTA vessel density maps of the retina were used to quantify vessel closure. Thickness of the ganglion cell + inner plexiform layer (GCL + IPL) was examined to identify retinal neurodegenerative changes. Diabetic retinopathy ETDRS classification was performed using the seven-field ETDRS protocol. **Results:** A total of 78 eyes/patients, aged 52 to 80 years, with T2D and ETDRS grades from 10 to 47 were followed for 3 years with annual examinations. A progressive increase in retinal vessel closure was observed. Vessel density (VD) showed higher decreases with retinopathy worsening demonstrated by step-changes in ETDRS severity scale (*p* < 0.001). No clear correlation was observed between neurodegenerative changes and retinopathy progression. **Conclusions:** Retinal vessel closure in NPDR correlates with DR severity progression. Our findings provide supporting evidence that OCTA metrics of vessel closure may be used as a surrogate for DR severity progression.

## 1. Introduction

Diabetic retinopathy (DR) is a major complication of type 2 diabetes (T2D) and a leading cause of visual impairment and blindness, and its incidence tends to increase [1]. 

Early and accurate identification of retinal changes and individual rates of progression are paramount to guide treatment decisions and determine prognosis and may help in the prevention of vision loss. In the initial stages, diabetic eyes may show neurodegeneration (ND), edema, and increases in vessel closure (VC). These changes occur to different degrees in different eyes, indicating that different mechanisms of retinal disease may predominate in different patients [2].

Our group has investigated DR biomarkers for progression profiles in different individuals, having proposed three phenotypes of progression [3,4], with different prognoses for progression to vision-threatening complications [5] and DR severity [6]. Indeed, the use of non-invasive imaging approaches has gained much relevance in the identification of DR biomarkers of staging and progression. Multimodal imaging approaches, such as OCT and OCT-Angiography (OCTA), facilitate the identification of different pathways of DR, namely ND, edema and ischemia [2,7]. Hence, OCTA has emerged as an innovative non-invasive tool to investigate quantitatively and qualitatively the retinal blood flow and capillary networks. It is a functional extension of structural OCT, that uses repeated B-scans to detect motion contrast, allowing the visualization of retinal microvasculature without intravenous dye injection [8]. Importantly, OCTA provides depth-resolved information on retinal circulation facilitating the evaluation of the individual’s retinal capillary plexus.

Taking advantage of OCTA, we have shown in a 2-year longitudinal study, that retinal VC increased with retinopathy progression, in contrast with edema and ND, which remained relatively stable [8]. There is, therefore, a clear need for evaluation of the OCTA metrics prognostic value in DR progression in longitudinal studies over longer periods. We present here a 3-year follow-up study, with data from a cohort of T2D individuals with non-proliferative diabetic retinopathy (NPDR), in which we have investigated the relationship between quantitative OCTA metrics, ND and DR severity progression.

## 2. Materials and Methods

This study is a 3-year prospective longitudinal study designed to analyze 90 eyes. The study was designed to analyze individuals with T2D and with NPDR (ETDRS grades 10 to 47), who have completed four visits in a period of 3-years of follow-up. The tenets of the Declaration of Helsinki were followed, approval was obtained from the AIBILI’S Ethics Committee for Health and written informed consent to participate in the study was obtained from all individuals after all procedures were explained.

Exclusion criteria included any previous laser treatment or intravitreal injections, presence of other retinal disease (e.g., age-related macular degeneration, glaucoma, or vitreomacular disease), high ametropia (spherical equivalent greater than −6 and +2 diopters), or any other systemic disease that could affect the eye, with special attention to uncontrolled systemic hypertension (values outside normal range: systolic 70–210 mmHg and diastolic 50–120 mmHg) and history of ischemic heart disease.

A total of 90 eyes of individuals with T2D and ETDRS levels between 10 and 47 were included, with a maximum glycated hemoglobinA_1c_ (HbA_1c_) value of 10%. A population of 84 eyes of individuals without diabetes, in the same age range, or other retinal diseases was used as a control group to set normal values and identify abnormal deviations between diabetic and non-diabetic control populations. No follow-up of non-diabetic individuals was performed. Age, duration of diabetes, HbA_1c_, and blood pressure levels were collected for each participant at the baseline visit.

All participants underwent a full ophthalmologic examination, including visual acuity, 7-field color fundus photography (CFP), SD-OCT, and OCTA imaging, at baseline and at the 1-year, 2-year and 3-year follow-up visits.

Eighty-four healthy control eyes (one eye per subject), from an age-matched population, were imaged in a single visit within the scope of the screening program, using SD-OCT and OCTA and were used as a reference control.

### 2.1. Seven-Field Color Fundus Photography

The 7-fields CFP were acquired using the Topcon TRC 50DX camera (Topcon Medical Systems, Tokyo, Japan), at 30/35°. The DR severity score was determined at baseline and at every annual visit by 2 independent graders in a context of an experienced reading center (Coimbra Ophthalmology Reading Center—CORC, Coimbra, Portugal) using a modified Airlie House classification scheme according to the ETDRS Protocol [9,10]. The observed agreement between the 2 graders was 97%. All disagreement cases were resolved by mutual agreement [11]. Step changes in the ETDRS retinopathy severity scale were used to describe worsening or improvement of the retinopathy [9,12].

### 2.2. Optical Coherence Tomography

Optical coherence tomography (OCT) was performed using the Cirrus HD-OCT 5000 (Carl Zeiss Meditec, Inc., Dublin, CA, USA). The Macular Cube 512 × 128 acquisition protocol, consisting of 128 B-scans with 512 A-scans each, was used to assess the subjects’ CRT. The average thickness values at the inner ring of the ganglion cell layer + inner plexiform layer (GCL + IPL) were gathered with Zeiss Cirrus standard reports.

Decreases in GLP + IPL were considered to identify ND, whereas CRT increases were considered to identify edema.

### 2.3. OCT-Angiography

OCT-Angiography data were collected by the Cirrus HD-OCT 5000 device using the Angiography 3 × 3 mm^2^ acquisition protocol, which consists of a set of 245 clusters of 4 B-scans repetitions, where each B-scan consists of 245 A-scans, over a 3 × 3 × 2 mm^3^ volume in the central macula.

The Carl Zeiss Meditec Density Exerciser (version:10.0.12787; Carl Zeiss Meditec, Inc., Dublin, CA, USA) was applied to calculate perfusion density (PD) and vessel density (VD) [8,13].

Vascular density metrics (VD and PD) for the Inner Ring region were analyzed for the SCP, DCP and FR. The circularity index of the Foveal Avascular Zone (FAZ) detected on the SCP was also computed. We also calculated the VD metrics of the FR, since those should be independent of projection artifacts that may affect the examination of the DCP.

All OCTA examinations underwent a quality check and normalization of signal strength as previously described [13,14]. Likewise, from 90 eyes included in this study, 13.3% were excluded as they did not meet the set of quality criteria in the first or last visits, leading to a final number of 78 eyes that allowed analysis of the 3-year progression. Included in the data set there is a small number of eyes with poor imaging quality in the second 13 (17%) and third visits 14 (18%) (Figure 1).

Capillary closure was identified by decreased VD or PD metrics measured in the SCP, DCP and FR.

### 2.4. Statistical Analysis

Variables were summarized for each diabetic retinopathy severity scale (DRSS) group, 10–20, 35, and 43–47, using mean and SD.

The χ^2^ test for categorical variables and the Kruskal–Wallis-H test for continuous variables were performed for comparison between the 3 ETDRS groups. To assess statistically significant differences between the measurements of healthy controls and each ETDRS group, the χ^2^ test was used for categorical variables and the Mann–Whitney *U* test was used for continuous variables.

For comparison of the progression of the disease in terms of VD and GCL + IPL layer Mann–Whitney U tests were used. Multiple regression analysis was performed to identify factors associated with VD. Correlation between changes of VD and GCL + IPL thickness from the first to the last visits was assessed by the Spearman’s rank correlation coefficient. Statistical analysis was performed with Stata 16.1 (StataCorp LLC, College Station, Texas, USA), and a *p* value ≤ 0.05 was considered statistically significant. When performing the comparison of VD metrics with ETDRS severity worsening, between the distinct ETDRS groups a Bonferroni correction was applied to correct for multiple comparisons.

## 3. Results

Of the 78 eyes analyzed, 24 (31%) were graded at ETDRS levels 10–20 at baseline (with 16 being level 10 and 8 level 20), 31 (40%) as ETDRS 35 and 23 as ETDRS 43–47 (29%). Demographic and baseline systemic and ocular parameters of 84 healthy control eyes and 78 T2D eyes included in the study are presented in Table 1. Of the systemic variables only HbA_1c_ shows statistically significant differences between the DR severity groups (*p* = 0.005). The ocular parameters, VD and PD, both representing VC (i.e., decreased perfusion of red blood cells) and FAZ circularity were significantly different between ETDRS groups, reflecting an association between retinal capillary non-perfusion and different severity grades of the retinopathy (Table 1). Across the different ETDRS levels examined, OCTA was able to detect differences in VC compared with changes in PD.

Presence of VC and ND varied widely within each of the three main ETDRS groups examined (Figure 1). There were eyes with definite VC and ND (≥2 standard deviation [2 SD] vs healthy controls) and eyes with minimal or no change.

The presence and degree of VC is shown to be related with the severity of the retinopathy as identified by ETDRS grade (Figure 1). At the last visit, decreases in VD larger than 1 or 2 SD in comparison with healthy controls were registered in 58% of the eyes in groups 10–20 (14 of 24), in 67% in group 35 (21 of 31) and 74% in group 43–47 (17 of 23). The presence of ND was detected less frequently. At the last visit, GCL + IPL thinning of more than 1 or 2 SD compared to healthy controls was identified in 33% of the eyes in group 10–20 (8 of 24), 69% in group 35 (20 of 31) and 52% in group 43-47 (12 of 23).

When analyzing only definite VC and ND (i.e., 2 SD changes related to normal healthy controls) in the 3-year follow-up visit, definite VC was identified in 33.3% of the eyes in group 10–20, 42% of the group 35 and in 47.8% of the group 43–47, with definite ND identified only in 8.3% of the eyes in group 10–20, 24.1% of the group 35 and 21.7% of the group 43–47 (Table 2).

Presence of definite VC and ND (≥2 SD changes) in the same eyes were not observed at the baseline visit in ETDRS group 10–20 but were present in 16% of the eyes in the last visit of the same group. In group 35, definite VC and ND were present in the same eyes in 9.6% at the baseline visit and 16% in the last visit. Finally, in group 43–47, definite VC and ND were present in the same eyes only in 8.7% of the eyes in both baseline and last visit, showing that dissociation between definite VC and definite ND predominates during the 3-year follow-up period (Table 2). Furthermore, no statistically significant correlations were found between the changes (first to last visits) of VD and GCL + IPL thickness, except for a weak correlation found on ETDRS 35 group when considering the DCP layer (ρ = 0.37, *p* = 0.045).

Concerning ETDRS level changes at 3 years of follow-up ten eyes (12.8%) presented a one-step improvement and 11 eyes showed worsening (14.1%), with one step-worsening found in 5 eyes (6.4%) and two-step worsening in the other 6 eyes (7.7%). When comparing VD between groups of ETDRS grade changes during the three-year follow-up period, significant differences were identified. The eyes with one-or two-step worsening showed a higher decrease in VD than the eyes that maintained their ETDRS grade or showed improvement during the three-year follow-up (Table 3). Comparison of VD Inner Ring metrics with disease progression showed statistical significance in all layers assessed, namely SCP, DCP and FR (respectively: *p* = 0.014, *p* = 0.048 and *p* = 0.047). No significant differences were observed in the GCL + IPL thinning between eyes that showed worsening of ETDRS severity grade and the eyes that maintained or improved during the 3-year period of study (Table 3).

When looking for factors associated with baseline VD on a univariate regression analysis, MAT and GCL+IPL were found to be significantly associated with VD (*p* = 0.003 and *p* = 0.011, respectively). These variables were then considered for a multivariate regression analysis along with demographic and systemic features. In this model, only MAT (ß = −0.096, 95%CI: −0.168 to −0.024, *p* = 0.009) and GCL + IPL (ß = 0.035, 95%CI: 0.003 to 0.067, *p* = 0.032) showed a statistically significant association with baseline VD.

## 4. Discussion

The results here reported confirm that eyes in the initial stages of retinopathy in T2D patients show evidence of VC and neurodegenerative changes and that these changes are present in different degrees in different patients even when classified as belonging to the same ETDRS severity grade. Moreover, the metrics of these changes show a wide range of values. Definite ND and VC (i.e., ≥2 SD changes) can occur very early in the disease process but are not present in every patient and, when present, they do not have the same rate of progression.

In this study, we have followed, for a period of 3 years, eyes categorized as no retinopathy, minimal, mild, and moderate retinopathy using seven-field ETDRS grading. Only VC (ischemia) showed significant progression during the 3-year period of follow-up. Furthermore, during this period, one- or two-steps worsening of retinopathy severity showed higher degrees in VD, confirming previous observations [8].

Our study confirms the presence of VC in the initial stages of DR, with earlier detection in the SCP, suggesting that one of the earliest changes associated with DR is reduced VD, possibly due to a decrease in retinal blood flow in selected capillaries [15,16]. The reduced capillary flow in the diabetic retina is most probably related to a decrease in the number of capillaries that carry red blood cells, instead of changes in capillary diameter, because skeletonized VD is the metric that better detects diabetic VC. The increase in VC due to the number of closed capillaries to red blood cell flow is compatible with the development of preferential channels or arteriovenous shunts which have been observed on histologic and trypsin-digest preparations of diabetic retinas [16,17,18]. We have reported VC results from SCP and DCP as well as from the full retina; although calculation of VD for full retina decreases the impact of projection artifacts, the overlap of vessels of SCP and DCP may affect its sensitivity.

Our study confirms that VC occurs initially in the perifoveal retina, progressing rapidly in the ETDRS levels 10, 20 and 35 but the changes in the perifovea appear to plateau at levels 43–47. This may be associated with a shift in the location of the VC to the midperiphery of the retina [19].

Our study shows that VC identified with OCTA reaches different degrees in patients with the same ETDRS severity grade and might be a tool to monitor DR progression. This observation suggests that different patients with T2D have different microvascular responses with some patients being able to maintain a viable retinal circulation and showing minimal changes, whereas others respond by poor capillary recruitment and progressive VC. Indeed, our results are in agreement with recent studies that suggest the predictive role of OCTA on DR progression and development of vision-threatening complications in diabetic individuals [20,21,22].

It is this variability of the retinal microvascular changes in T2D, regarding both their initiation and progression in relatively initial stages of NPDR that we consider a most relevant finding of this study. Some patients show steady and progressive worsening whereas others show a variable course and evidence of reversibility of their changes, as well demonstrated in Figure 1. These observations offer two important messages. First, the reversibility of the VC opens the door for early intervention with the possibility of stopping disease progression. Second, each patient should be followed closely, and a variety of risk factors should be considered to determine a specific risk profile for that patient. It demonstrates the complexity of diabetic retinal disease and indicates that multiple genes and environment factors may be involved, creating different subtypes of progression.

Neurodegeneration identified by GCL + IPL thinning has been proposed as playing a major role in the development and progression of DR. This study shows that ND association with VC was present at baseline in this cohort, but this association appears to uncouple as the disease progresses, as shown in Figure 1.

A strict quality check of OCTA vessel metrics is particularly important and necessary when comparing different examinations performed in the same patient in longitudinal studies and to identify disease progression. In this study, with examinations performed by experienced technicians, data from 17% of the examinations did not pass the final quality check and had to be excluded from the data analysis.

A limitation of this study is the number of eyes included in the study. Other limitations include the lack of use of a projection removal algorithm that can increase sensitivity, and the limited scan field (3 × 3 mm), which prevents a more detailed overview of the VC differences between ETDRS levels observed with Swept-Source OCTA (15 × 9 mm) [13]. However, measurements of retinal thinning were performed in retinas that remained structurally preserved with no evidence of cystoid changes. Of special value, a strict quality check was performed by a masked grader and normalization of OCTA metrics based on signal strength was performed.

In conclusion, OCT-angiography metrics of retinal VC, more specifically VD measurements based on skeletonized images, obtained in a noninvasive manner that allow repeated examinations and close follow-up, are particularly promising candidates as biomarkers of DR severity progression and are expected to impact disease management.

## 5. Conclusions

Eyes with initial stages of retinopathy in T2D individuals followed by OCT and OCTA during a 3-year period demonstrate progressive increase in vessel closure. Vessel density showed higher decrease in eyes with retinopathy worsening demonstrated by step changes in ETDRS severity scale. Neurodegenerative changes, although associated with vessel closure at baseline, did not uniformly progress during the 3-year period of follow up.

## Figures and Tables

**Figure 1 jcm-10-02296-f001:**
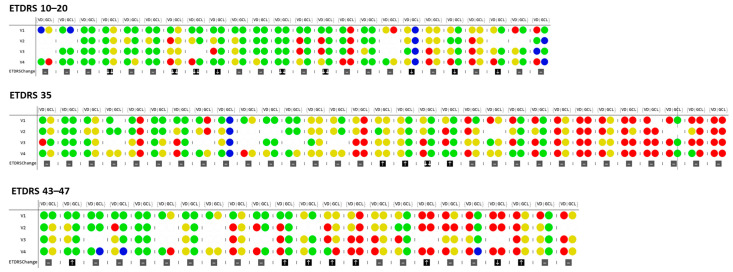
Schematic representation of individual vessel density values in the SCP inner ring and thinning of GCL + IPL and its progression over the four visits, presented according to differences and variation in VD across ETDRS groups. The values are given in relation to the control group: Values within a normal range are depicted in green; 2 SD decrease is depicted in red; 1 SD decrease is depicted in yellow; and 2 SD increase are shown in blue. Circles without color indicate that reliable measurements could not be obtained in that specific visit due to insufficient image quality. Arrows indicate ETDRS step progression: one or two increase (↑), maintenance (-) or decrease (↓). VD: Vessel density; GCL: Ganglion cell layer and Inner plexiform layers. V1: Baseline visit; V2: 1 year visit; V3: 2-year visit; V4: Last visit (3 years).

**Table 1 jcm-10-02296-t001:** Baseline characteristics for healthy controls and T2D individuals considering distinct DRSS stages of the disease.

	Healthy Controls	ETDRS 10–20	ETDRS 35	ETDRS 43–47	*p*-Value (Between the Three ETDRS Groups) **
	(*n* = 84)	(*n* = 24)	(*n* = 31)	(*n* = 23)	
Sex, Male/female	39/45	16/8	25/6	17/6	*0.499*
		*0.08*	***0.001***	***0.019***	
Age, years	69.2 ± 4.5	69.5 ± 5.9	65.4 ± 5.5	66.5 ± 7.2	*0.064*
		*0.706*	***0.002***	*0.213*	
Diabetes duration, years	---	18.2 ± 7.1	16.5 ± 6.8	17.6 ± 5.8	*0.577*
BCVA, letters	---	85.5 ± 4.1	85.3 ± 4.4	86.8 ± 3.4	*0.528*
HbA1c, %	---	6.9 ± 1.1	7.3 ± 1.1	8.0 ± 1.2	***0.005***
VD, SCP, inner ring, mm^−1^ *^(p-value *)^*	22.3 ± 0.89	21.7 ± 1.1	20.9 ± 1.1	21.2 ± 1.3	***0.033***
		0.058	***p*** ** < 0.001**	***p*** ** < 0.001**	
VD, DCP, inner ring, mm^−1^ *^(p-value *)^*	17.0 ± 2.14	*17.2 ± 1.97*	*16.2 ± 2.2*	*16.4 ± 2.2*	*0.324*
		*0.918*	*0.078*	*0.310*	
VD, FR, inner ring, mm^−1^ *^(p-value *)^*	23.7 ± 0.90	23.5 ± 1.1	22.6 ± 1.1	22.9 ± 1.1	***0.009***
		*0.958*	***p*** ** < 0.001**	***0.010***	
PD, SCP, inner ring, mm^−1^ *^(p-value *)^*	0.398 ± 0.02	0.403 ± 0.02	0.390 ± 0.02	0.400 ± 0.02	***0.038***
		*0.416*	***0.006***	*0.073*	
PD, DCP, inner ring, mm^−1^ *^(p-value *)^*	0.315 ± 0.04	0.332 ± 0.03	0.312 ± 0.04	0.318 ± 0.04	*0.221*
		*0.535*	*0.119*	*0.491*	
PD, FR, inner ring, mm^−1^ *^(p-value *)^*	0.416 ± 0.02	0.429 ± 0.02	0.414 ± 0.02	0.426 ± 0.02	***0.011***
		*0.051*	*0.059*	*0.176*	
FAZ circularity index *^(p-value *)^*	0.687 ± 0.07	0.638 ± 0.2	0.607 ± 0.1	0.560 ± 0.1	***0.004***
		*0.636*	*0.205*	***p*** ** < 0.001**	
GCL + IPL, Inner Ring, µm (58 healthy controls) *^(p-value *)^*	82.7 ± 5.5	82.8 ± 9.1	77.6 ± 8.1	78 ± 6.8	*0.087*
		*0.710*	***0.003***	***0.007***	

Data presented as Mean ± SD. * χ² test (categorical variables) and Mann–Whitney U-test (continuous variables) for comparison between Healthy Controls and each ETDRS group. ** χ² test (categorical variables) and Kruskal–Wallis test (continuous variables) for comparison between the three ETDRS groups. BCVA: Best corrected visual acuity; HbA1c: Glycated hemoglobin; VD: Vessel density; PD: Perfusion density; SCP: Superficial capillary plexus; DCP: Deep capillary plexus; FR: Full retina; FAZ: Foveal avascular zone; GCL + IPL: Ganglion cell + Inner plexiform layer.

**Table 2 jcm-10-02296-t002:** Percentage and correlation of changes of vessel closure and neurodegenerative changes ≥ 2 SD at baseline and last visit.

≥2D Changes	ETDRS 10–20 (*n* = 24)	ETDRS 35 * (*n* = 31)	ETDRS 43–47 (*n* = 23)
***Visit 1***			
Vessel closure (VC)	8.3%	38.7%	30.4%
Neurodegeneration (ND)	4.2%	22.6%	13.0%
VC and ND in the same eye	0.0%	9.6%	8.7%
***Visit 4***			
Vessel closure (VC)	33.3%	41.9%	47.8%
Neurodegeneration (ND)	8.3%	24.1%	21.7%
VC and ND in the same eye	16.0%	16.1%	8.7%
VD (SCP) and GCL + IPL thickness correlation change (V4 − V1)	0.07 (*p* = 0.739)	0.03 (*p* = 0.872)	−0.25 (*p* = 0.246)

Percentage of eyes presenting 2 SD decreases in vessel closure (vessel density decrease) and neurodegeneration (GCL + IPL thinning), relative to normal healthy control eyes in the baseline visit (V1) and last visit of the 3-year follow-up (V4), in ETDRS groups 10–20, 35 and 43–47. * ETDRS group 35 has *n* = 29 for eyes assessed for neurodegeneration. Spearman’s rank correlation coefficients between the change (V4–V1). VC: Vessel closure, ND: Neurodegeneration.

**Table 3 jcm-10-02296-t003:** Vessel density and ganglion cell layer + inner plexiform layer thickness comparison between T2D individuals that improved and worsened 1 and 2 steps in DRSS stage, at the end of the three-year follow-up.

ETDRS Change		Vessel Density Inner Ring mm^−1^						Layer Thickness Inner Ring, µm	
		SCP		DCP		FR		GCL + IPL	
		V1	V4 − V1	V1	V4 − V1	V1	V4 − V1	V1	V4 − V1
**Worsening (1 and 2 steps****)** (*n* = 11)	AVG	21.5	−1.3	16.7	−2.1	23.1	−1.3	81.9	−0.5
	SD	0.9	1.0	1.7	1.5	0.9	1.1	8.7	1.2
	Min	20.0	−2.9	14.6	−4.8	21.3	−3.2	65.0	−2.0
	Max	22.5	−0.3	19.7	−0.1	24.3	−0.1	98.0	1.0
**No change + improving** (*n* = 67)	AVG	21.2	−0.4	16.6	−1.0	23.0	−0.4	78.9	−1.1
	SD	1.2	1.0	2.2	1.7	1.2	1.0	8.3	5.6
	Min	18.5	−1.8	11.4	−4.4	20.3	−1.9	58.0	−26.0
	Max	24.9	3.1	22.1	2.9	26.5	3.1	110.0	18.0
**Statistical Diff V4** **− V1 (*p*-value)**		**0.014 ***		**0.048**		**0.047**		0.810	

Vessel density (mm^−1^) for the Inner Ring were collected using the Cirrus HD-OCT 5000 with AngioPlex module. The average thickness values at the inner ring of the GCL + IPL were collected from Zeiss Cirrus standard reports. Comparison was performed by Mann–Whitney U-test, followed by Bonferroni correction for multiple comparisons. Significant *p*-values (*p* < 0.005) are highlighted in bold. * indicates a significance level of 0.017 obtained after Bonferroni correction. AVG: Average; SD: Standard deviation; Min: Minimum value; Max: Maximum value; SCP: Superficial capillary plexus; DCP: Deep capillary plexus; FR: Full retina; GCL + IPL: Ganglion cell + Inner plexiform layers. V1: Baseline visit; V4: Last visit (3 years).

## Data Availability

Data will be available upon request to correspondent author.

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
