# Peer review of "Optical Coherence Tomography Angiography Metrics Monitor Severity Progression of Diabetic Retinopathy—3-Year Longitudinal Study"

_jcm, 2021, doi:10.3390/jcm10112296_

Round 1
Reviewer 1 Report
Comment #1: In this study, many important data about systemic condition, i.e., HbA1c, eGFR, systemic blood pressure, were lacking. These systemic factors may be involved with the progression of vascular closure. Especially, uncontrolled systemic hypertension may have some influences on vascular status in diabetic patients.
Comment #2: If macular edema occurred during 3 years follow-up period, it may affect the accuracy of OCTA findings. How many patients had macular edema during follow-up period.
In addition, were there any patients with the anti-VEGF treatment for diabetic macular edema?
Comment #3: In addition, the progression of cataract may have some influences on the qualities of OCTA analysis during the follow-up period. Please provide the data about the status of the lens. In figure 1, there were some patients in which the reliability of OCTA qualities changes during follow-up period. Some patients may had cataract surgery.
Comment #4. Please provide the representative cases with the 1-step improve, 1-step worsening, and 2-step worsening of diabetic retinopathy during 3 years.
Comment #5: In table 4, there we no information about microaneurysm in this study. Therefore, the comparison of phenotype A and B with C seems to be very confusing.
Author Response
Reviewers’ comments
Dear Dr Stefan Nedelcu
Thank you for allowing us to resubmit a revised version of the manuscript, with the corrections and suggestions requested by the reviewers, which have contributed to an improved manuscript.
Alterations in the manuscript are highlighted with track changes.
Reviewer #1
Comment #1: In this study, many important data about systemic condition, i.e., HbA1c, eGFR, systemic blood pressure, were lacking. These systemic factors may be involved with the progression of vascular closure. Especially, uncontrolled systemic hypertension may have some influences on vascular status in diabetic patients.
As stated in line 86, uncontrolled hypertension was an exclusion criteria. Regarding other systemic factors assessed, only diabetes duration and HbA1c showed associations with vascular status of our diabetic population.
Comment #2: If macular edema occurred during 3 years follow-up period, it may affect the accuracy of OCTA findings. How many patients had macular edema during follow-up period.
There where no patients with cystoid macular edema, which could affect the accuracy of OCTA findings.
In addition, were there any patients with the anti-VEGF treatment for diabetic macular edema?
No, Anti-VEGF treatment (intravitreal injections) was also an exclusion criterion, as stated in line 79.
Comment #3: In addition, the progression of cataract may have some influences on the qualities of OCTA analysis during the follow-up period. Please provide the data about the status of the lens. In figure 1, there were some patients in which the reliability of OCTA qualities changes during follow-up period. Some patients may had cataract surgery.
No patients in the study had any eye surgery during the 3-year follow-up period.
Comment #4. Please provide the representative cases with the 1-step improve, 1-step worsening, and 2-step worsening of diabetic retinopathy during 3 years.
This information is now included in figure 1.
Comment #5: In table 4, there we no information about microaneurysm in this study. Therefore, the comparison of phenotype A and B with C seems to be very confusing.
The information on phenotypes has been removed at the suggestion of another reviewer.

Reviewer 2 Report
Marques et al. present an incremental and expanded analysis of eye with NPDR followed with spectral domain-optical coherence tomography (SD-20 OCT) and OCT-Angiography (OCTA) assessing structural and vascular features of retinopathy over a three year period. The study represents a great deal of work. A valuable data set that has been collected, containing a wealth of information. Some of this work has been previously published, albeit in more limited form and scope compared with the present study.
My enthusiasm for this effort is tempered by the persistence in grouping from the outset patients into phenotypes "A,B,C". This is not commonly accepted nomenclature for classification. Though novel, the analysis would be better served by first examining this small cohort of eyes as a group to look for patterns associated with ND and VC/VD before splitting eyes into these subcategories. Even dividing eyes by ETDRS levels, though well established to correlate with DR progression and complications, may diminish the ability to understand patterns in the data when analysis is compartmentalized.
Abstract / Methods
Consider revising lines 17-21 to read:
Three-year prospective longitudinal observational cohort study of patients with type 2 diabetes (T2D) using spectral domain-optical coherence tomography (SD-OCT) and OCT-Angiography (OCTA). The study eye from each patient was examined [four times] at one-year intervals.
This cleanly makes clear that you have just one eye per patient in the study
Reference to "eyes/patients" is distracting.
Line 90 ...at [the/their] baseline visit.
Line 91 ...All participants underwent full ophthalmologic examination, including visual acuity, 7-field color fundus photography (CFP), SD-OCT, and OCTA imaging [at baseline and at the] 1-year, 2-year and 3-year [follow-up] visits.
Line 115 Phenotype Classification appears to be at 6 months, but the full ophthalmologic examinations were only performed at 1-year intervals. These additional study visits should be made clear.
There are 90 participants in the study, but only 78 eyes from 78 participants were included, one eye per subject. This could be made more clear.
The methods section is too long. The extension of previous work is logical and admirable, but it feels wholly unnecessary to repeat all of these methods. As an alternative, a large portion of this text could be moved to a supplement.
This is a follow up study. Methods should not be repeated unless they differ from prior published reports. For example, lines 144-148 could be streamlined to read:
"All OCTA examinations underwent a quality check and normalization of signal strength as previously described[17]."
Results
Line 184-185
Consider restating:
Across the different ETDRS levels examined, OCTA was able to detect differences in VC compared with changes in PD.
Figure 1
Clustering (ordering) patients by the pattern and severity of changes could allow for a more effective visual demonstration of the differences and variation across these ETDRS cutoffs. Group the green patients together with graduations up to the red. Consider placing each ETDRS cohort on a separate line rather than running them together like a strip wrapping along the page.
Given that there were eyes whose ETDRS score changed over the course of the 3 year study, which ETDRS level is used to group the eyes in this figure? It should be stated, ETDRS Level at baseline. Perhaps the figure could be regenerated using the final ETDRS level at end-of-study to see if there is more of a pattern apparent in the data. Clearly this figure and Table 2 support the conclusion that there is not a statistically significant progression of consistent ND changes in these eyes based on how they have been classified in the study (ETDRS level).
Table 3 Bonferroni correction is only listed (applied) to the SCP. I think it makes sense to simply present the p-value taking the Bonferroni correction for each of the groups for which it is applicable. There is no need to list two competing p-values. When the Bonferroni correction is appropriate, simply apply it.
Phenotypes "A,B,C" are not commonly accepted nomenclature and should be avoided in the manuscript where possible to save the reader from having to go back to the definition of these classifications.
I believe that analysis of progression of ND VC and ETDRS stage should be performed for all patients together rather than splitting patients into separate groups, pre-defined as Phenotypes "A,B,C". As mentioned, patients vary widely with regards to BOTH ND and VD/VC metrics and an analysis of these metrics on a continue scale should be undertaken. As it is, the authors choose to group Phenotypes A & B together for their analysis, I suspect because the comparison across these three groups did not reach significance individually, but this is not presented.
Line 292 word choice: consider "plateau" rather than "stabilize"
Line 303: Consider revision. It is not so much a "response" of the vascular but a changes in the vasculature.
Line 310-311: The complexity of diabetic retinal disease does not necessarily imply that multiple genes may be involved. It could just as easily be the wide variety of environmental factors at play, or both genes and environment.
Line 315: By "irregular" do you mean random, or better stated unassociated or uncoupled with ETDRS stage?
Line 331-334: This conclusion does not make sense!
"In conclusion, eyes with no retinopathy, minimal, mild or moderate NPDR followed, for a period of 3 years, with annual visits, show evidence of ND and ischemia distributed over a wide range of values in different patients and that reversibility of these changes may occur in individual patients."
All eyes had retinopathy, not "no retinopathy".
Line 340-345: The second "conclusion" section is better stated. I suggest cutting the previous conclusion state above.
Line 345: Consider revision: ... [did not uniformly progress] during the 3-year period of follow up.
Author Response
Reviewers’ comments
Dear Dr Stefan Nedelcu
Thank you for allowing us to resubmit a revised version of the manuscript, with the corrections and suggestions requested by the reviewers, which have contributed to an improved manuscript.
Alterations in the manuscript are highlighted with track changes.
Reviewer #2
Marques et al. present an incremental and expanded analysis of eye with NPDR followed with spectral domain-optical coherence tomography (SD-20 OCT) and OCT-Angiography (OCTA) assessing structural and vascular features of retinopathy over a three year period. The study represents a great deal of work. A valuable data set that has been collected, containing a wealth of information. Some of this work has been previously published, albeit in more limited form and scope compared with the present study.
My enthusiasm for this effort is tempered by the persistence in grouping from the outset patients into phenotypes "A,B,C". This is not commonly accepted nomenclature for classification. Though novel, the analysis would be better served by first examining this small cohort of eyes as a group to look for patterns associated with ND and VC/VD before splitting eyes into these subcategories. Even dividing eyes by ETDRS levels, though well established to correlate with DR progression and complications, may diminish the ability to understand patterns in the data when analysis is compartmentalized.
We agree with the reviewer comments and views. Including the Phenotypes, A, B and C confuses the messages and therefore we have taken out this information from this contribution. We thank the reviewer for his suggestion.
1.Abstract / Methods
Consider revising lines 17-21 to read:
Three-year prospective longitudinal observational cohort study of patients with type 2 diabetes (T2D) using spectral domain-optical coherence tomography (SD-OCT) and OCT-Angiography (OCTA). The study eye from each patient was examined [four times] at one-year intervals.
This cleanly makes clear that you have just one eye per patient in the study. Reference to "eyes/patients" is distracting.
Thank you. We have revised the sentence to clarify the use only one eye per patient “Three-year prospective longitudinal observational cohort of individuals with type 2 diabetes (T2D), one eye per person, using spectral domain-optical coherence tomography (SD-OCT) and OCT-Angiography (OCTA).” (lines 19-21)
This was also clarified throughout the manuscript.
- Line 90 ...at [the/their] baseline visit.
Thank you for your correction. We have included it.
3.Line 91 ...All participants underwent full ophthalmologic examination, including visual acuity, 7-field color fundus photography (CFP), SD-OCT, and OCTA imaging [at baseline and at the] 1-year, 2-year and 3-year [follow-up] visits.
Thank you for your suggestion. It is now included.
- 4. Line 115 Phenotype Classification appears to be at 6 months, but the full ophthalmologic examinations were only performed at 1-year intervals. These additional study visits should be made clear.
As suggested, we have removed the phenotype analysis from this work, so we also excluded this part of the methods section.
- There are 90 participants in the study, but only 78 eyes from 78 participants were included, one eye per subject. This could be made more clear.
As stated in lines 150-151, from the 90 eyes enrolled in this study, 13.3% were excluded as they did not meet the set quality criteria in the first or last visits, leading to a final number of 78 eyes that allowed analysis of the 3-year progression. These criteria were based in a quality check to discard acquisitions having a Signal Strength (SS) lower than 7, motion artifacts or evidence of defocus or blur in more than 25% of the area under analysis, as stated in lines (as previously stated), now in line 145.
- The methods section is too long. The extension of previous work is logical and admirable, but it feels wholly unnecessary to repeat all of these methods. As an alternative, a large portion of this text could be moved to a supplement.
We have now removed the phenotype classification form the methods section. We have also shortened the OCTA section. Other methodologies, that might seem extensive, namely Study Design and OCTA, are of major relevance and we believe that they should be presented. We have tried to shorten this section.
- This is a follow up study. Methods should not be repeated unless they differ from prior published reports. For example, lines 144-148 could be streamlined to read:
"All OCTA examinations underwent a quality check and normalization of signal strength as previously described[17]."
Thank you for your suggestion, we have modified the manuscript as suggested. (lines 145-146)
Results
- 8. Line 184-185. Consider restating:
Across the different ETDRS levels examined, OCTA was able to detect differences in VC compared with changes in PD.
Thank you for your suggestion, this sentence was restated.
- Figure 1:
Clustering (ordering) patients by the pattern and severity of changes could allow for a more effective visual demonstration of the differences and variation across these ETDRS cutoffs. Group the green patients together with graduations up to the red. Consider placing each ETDRS cohort on a separate line rather than running them together like a strip wrapping along the page.
We have modified figure 1 as suggested.
Given that there were eyes whose ETDRS score changed over the course of the 3 year study, which ETDRS level is used to group the eyes in this figure? It should be stated, ETDRS Level at baseline. Perhaps the figure could be regenerated using the final ETDRS level at end-of-study to see if there is more of a pattern apparent in the data. Clearly this figure and Table 2 support the conclusion that there is not a statistically significant progression of consistent ND changes in these eyes based on how they have been classified in the study (ETDRS level).
We have now included the step changes in each patient in figure 1
- Table 3 Bonferroni correction is only listed (applied) to the SCP. I think it makes sense to simply present the p-value taking the Bonferroni correction for each of the groups for which it is applicable. There is no need to list two competing p-values. When the Bonferroni correction is appropriate, simply apply it.
Bonferroni correction was applied SCP, DCP, FR and GCL+IPL, but only maintained statistical significance in the SCP after Bonferroni correction for multiple comparisons. This was clarified in the table legend.
12.Phenotypes "A,B,C" are not commonly accepted nomenclature and should be avoided in the manuscript where possible to save the reader from having to go back to the definition of these classifications.
I believe that analysis of progression of ND VC and ETDRS stage should be performed for all patients together rather than splitting patients into separate groups, pre-defined as Phenotypes "A,B,C". As mentioned, patients vary widely with regards to BOTH ND and VD/VC metrics and an analysis of these metrics on a continue scale should be undertaken. As it is, the authors choose to group Phenotypes A & B together for their analysis, I suspect because the comparison across these three groups did not reach significance individually, but this is not presented.
We have eliminated the information on the phenotypes A, B and C from this manuscript (methods section, results section (lines 256-260), table 4 and in the discussion section (lines 287-290)
- Line 292 word choice: consider "plateau" rather than "stabilize"
Thank you, we have replaced it.
- Line 303: Consider revision. It is not so much a "response" of the vascular but a changes in the vasculature.
Thank you, we have replaced the word “response” by “changes” as suggested.
- Line 310-311: The complexity of diabetic retinal disease does not necessarily imply that multiple genes may be involved. It could just as easily be the wide variety of environmental factors at play, or both genes and environment.
Thank you, we have included it
- Line 315: By "irregular" do you mean random, or better stated unassociated or uncoupled with ETDRS stage?
As suggested, we have modified the text, which is now stating “This study shows that ND association with VC was present at baseline in this cohort, but this association is appears to uncouple as disease progresses” (lines 326-330)
17.Line 331-334: This conclusion does not make sense!
"In conclusion, eyes with no retinopathy, minimal, mild or moderate NPDR followed, for a period of 3 years, with annual visits, show evidence of ND and ischemia distributed over a wide range of values in different patients and that reversibility of these changes may occur in individual patients."
All eyes had retinopathy, not "no retinopathy".
We agree with the reviewer and we have removed this sentence.
18.Line 340-345: The second "conclusion" section is better stated. I suggest cutting the previous conclusion state above.
Thank you, we have changed the manuscript according to your suggestion.
- Line 345: Consider revision: ... [did not uniformly progress] during the 3-year period of follow up.
We also accepted your suggestion. Thank you.

Round 2
Reviewer 2 Report
The paper has been much improved, but the quality of the writing and presentation (organization) has changed less than I had hoped.
I believe that the analysis is still overly compartmentalized. Dividing a small number of eyes by ETDRS stages reduces the statistical power of the study.
The origin of the control eyes, and number of times imaged, is NOT in the methods. It may also be a point of possible bias if more than one eye per control subject is used. This needs to be spelled out in the methods
Abbreviation definitions are missing from some tables , eg “V1”.

Author Response
Dear Dr Stefan Nedelcu
Thank you for allowing us to resubmit a revised version of the manuscript, with the corrections and suggestions requested by the reviewer 2, which have contributed to an improved manuscript.
Alterations in the manuscript are highlighted with track changes.
Reviewer #2
Comments and Suggestions for Authors
The paper has been much improved, but the quality of the writing and presentation (organization) has changed less than I had hoped.
I believe that the analysis is still overly compartmentalized. Dividing a small number of eyes by ETDRS stages reduces the statistical power of the study.
We believe that the results need to be compartmentalized in ETDRS stages in order to address the differences analyzed regarding severity progression in Diabetic Retinopathy.
The origin of the control eyes, and number of times imaged, is NOT in the methods. It may also be a point of possible bias if more than one eye per control subject is used. This needs to be spelled out in the methods.
This information was now included in the methods section lines (86-87) “Eighty-four healthy control eyes (one eye per subject), from an age-matched population, were imaged in a single visit within the scope of a screening program, using SD-OCT and OCTA and used as reference control.”
Abbreviation definitions are missing from some tables, eg “V1”.
We have now reviewed all definitions used in the tables.
This manuscript is a resubmission of an earlier submission. The following is a list of the peer review reports and author responses from that submission.